# A Self-Supervised Pre-Training Model for Time Series Classification based on Data pre-processing

## Abstract

Currently, time series is widely used in the industrial field. Many scholars have conducted research and made great progress, including pre-training models. By training the model with a large amount of data similar to a certain field, and then fine-tuning the model with a small amount of samples, a high-precision model can be obtained, which is of great value in the industrial field. However, there are two main problems with current models. First, most of them use supervised classification. Although the accuracy is high, it is not practical for many real-world data with few labeled samples. Secondly, most researchers have recently focused on contrastive learning, which has higher requirements for the form and regularity of data, indicating that they have not targeted these issues. To solve these problems, we propose an self-supervised pre-processing classification model for time series classification. First, according to the inherent features of the data, the way of data pre-processing is determined by judging the attributes of the time series. Second, a sorting similarity method is proposed for contrastive learning, and a rough similarity is used in the pre-training stage, while our sorting loss function is used in the fine-tuning stage to improve overall performance. After that, extensive experiments were conducted on 8 different real-world datasets from various fields to verify the effectiveness and efficiency of the proposed method.

## 1 Introduction

The extensive use of time series in the industrial field (Bi et al., 2023)(Li et al., 2020)(Gupta et al., 2020) is beyond doubt. For example, monitoring scenarios such as earthquakes, tsunamis, and bridge construction sites and high-altitude operations require a large number of sensors for the monitoring of the construction process. The data generated during these practical application scenarios are extremely valuable and rare (Narwariya et al., 2020). At the same time, many biological signals, such as electroencephalography (EEG) (Zhang et al., 2021) and electromyography (EMG) (Sahu et al., 2023), although there are many disease signals, still require targeted treatment. Therefore, if we can train a classification model based on a small amount of sample data, an accurate classification of domain data with high sample acquisition costs and limited small samples becomes meaningful and valuable time can also be saved. figure 1 shows the entire process of a microseismic signal from generation to reception. The time difference in the signal received by the sensor is likely caused by the complex geological environment below. In this case, although it is the same signal source, the form of the time series may be completely different, which greatly affects the classification of the time series.

Currently, there have been many research achievements in the field of natural language processing (Tinn et al., 2023)(Antonello et al., 2020) and the digital image processing (Chen et al., 2020)(Yuan & Lin, 2021) by pre-training a classification model for sequential and fine-tuning it on a small sample. In recent years, a large number of researchers have been attracted to the field of time series, such as pretraining models (Kashiparekh et al., 2019)(Malhotra et al., 2017)(Zheng et al., 2022), transfer learning (Ismail Fawaz et al., 2018a)(Laptev et al., 2018)(Kimura et al., 2020), model fine-tuning (Gong et al., 2023)(Qian et al., 2021)(Asadi & Regan, 2020), data augmentation (Yang et al., 2023)(Wen et al., 2020)(Ismail Fawaz et al., 2018b), etc. In transfer learning, most of the work is achieved by transferring model weight parameters and domain invariants. However, for actual data,

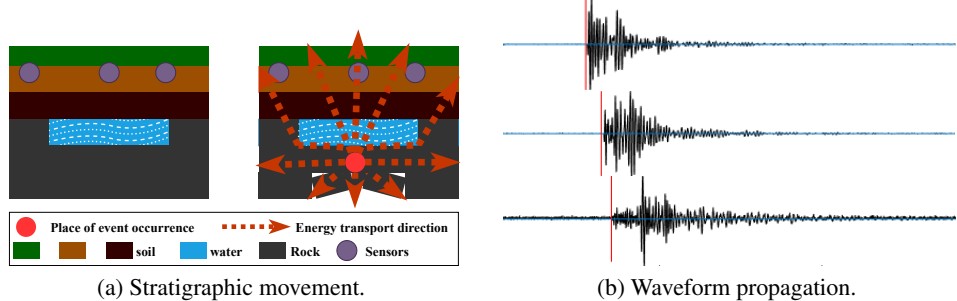

(a) Stratigraphic movement.      (b) Waveform propagation.

Figure 1: A toy example of micro-earthquakes.

the transfer of model weights requires researchers to have relevant experience in model training, which is difficult for non-professional researchers. In addition to the above two methods, model fine-tuning is also a hot topic in recent years. Since the concept of fine-tuning (Hinton & Salakhutdinov, 2006) was proposed, great progress has been made. Although it does not require strong computing power, there may be small sample biases in fine-tuning itself, which raises questions about whether fine-tuning is really effective.

Unlike transfer learning, the pre-training model proposed in this paper does not require a large amount of pre-training data. In the pre-training stage, contrastive learning is used (Tonekaboni et al., 2021)(Franceschi et al., 2019)(Yang & Hong, 2022), which does not require the data to have labels. At the fine-tuning stage, a small amount of labeled data is required and we use a novel similarity method to enhance performance. Following the work of Zhang et al. (2022), we propose a novel method of data pre-processing and a new similarity calculation function for the fine-tuning stage. The proposed model was extensively tested, and the contributions are as follows.

• A platform filtering method and a self-adaptive FIR filter is proposed, which preprocesses data according to its morphological characteristics and physical meaning to reduce the amount of invalid data. At the same time, image-like translation and flipping augmentations are applied to the data to improve the accuracy of pre-training models.

• Based on the loss of NT-Xent in the past, a new sorting similarity is proposed to fine-tune time series replacing cosine similarity. Meanwhile, a data pre-process time series classification model (DPTSC) is proposed to weaken the impact of mutations on the data.

• Extensive experiments on $4$ groups of $8$ real data sets show that our proposed method has better accuracy, precision, recall, F1 score and AUROC and AUPRC, than the state-of-art.

## 2 RELATED WORKS

### 2.1 SUPERVISED AND SEMI-SUPERVISED LEARNING OF TIME SERIES CLASSIFICATION METHOD

Xi et al. (2023) proposed an LB-SimTSC method, they suppose that each sample in the batch should be similar to each other and construct a graph with DTW distance, but they neglected the fact that the same class has different shapes, DTW cannot obtain the more information of curve. Xi et al. (2022) proposed a self-supervised module that divided the time series into three segments and combined them in different ways to form positive and negative samples, assisting the supervised classification module of Time Series Classification (TSC). However, due to the issue of the rationality of segment combinations during classification, the self-supervised module had certain limitations. Multivariate time series involves merging the features of different types of sequential data for classification, fully utilizing the gain information. Karim et al. (2019) proposed a deep learning classification method for multivariate time series, which utilized squeeze-and-excite blocks to extract high-dimensional information. However, due to the large amount of multivariate time series data, the training time was long. Wang et al. (2017) proposed an end-to-end FCN and ResNet network architecture, and end-to-end training has recently begun to receive attention. Similar work pursued modifications

to the network architecture, believing that complex pre-processing of data itself is a redundant operation. Unlike images, time series itself has data offsets, making it difficult to extract high-level features. Wei et al. (2023) proposed a method for extracting time-frequency features from unlabeled data, using time-frequency features to enhance the high-dimensional features of unlabeled data, and proposing an MTL framework that uses unsupervised data to provide curve knowledge and supervised classification features from labeled data. However, relative to the small amount of labeled data, this semi-supervised method relies heavily on the feature consistency of intra-class data.

## 2.2 SELF-SUPERVISED CONTRASTIVE LEARNING FOR TIME SERIES CLASSIFICATION.

Zhang et al. (2022) proposed a method of pretraining on a certain dataset and fully fine-tuning on a similar dataset, which achieved high accuracy, but some of the datasets lacked data pre-processing. Jia et al. (2021) proposed the use of brain functional graphs and brain functional region distance graphs. It also introduces an adaptive training method to fuse them into high-dimensional features. By applying adversarial domain generalization, the graph model is modified to achieve higher accuracy. However, a drawback is that it calculates the maximum distance between two waveforms without fully utilizing the temporal features. As a result, if the high-dimensional features cannot adequately represent the current class, this significantly affects accuracy. Lu et al. (2022) proposed to learn the feature distribution of time series beyond the out-of-distribution and use adversarial networks to learn this potential distribution, which preserves both diversity and consistency. However, the label obtained by the feature extraction function may not represent the high-dimensional features of the current data and may have the opposite effect. Eldele et al. (2021) proposed a double augmentation strategy of data and a non-supervised self-learning contrastive learning method that can characterize time series. The enhancement strategy is achieved through random jittering of time series, so selecting the parameters is important. Jawed et al. (2020) proposed a self-supervised auxiliary network that uses the features of unlabeled data to distinguish each other, which is equivalent to pseudo-labeling, and proposed a multitask learning method. However, it still has certain limitations in the division of feature weights in multitask learning. The above works have conducted in-depth research on supervised, semi-supervised, and self-supervised learning in time series representation learning and classification. However, there are still issues with intra-class offsets and non-standard data in time series classification, and different experts have inconsistent views. There are also problems with incorrect labeling, so there are still limitations in time series classification.

## 3 PRELIMINARIES

**Pre-training Dataset** Given a set of time series that needs to be pre-trained, denoted as $\mathbb{T} = \{\boldsymbol{x}_1, \ldots, \boldsymbol{x}_p\}$ contains $p$ samples. Each sample consists of $C_{\mathbb{T}}$ channels, and each channel contains $|\boldsymbol{x}_i|$ data points. The pre-training data does not include any labels $l_{\mathbb{T}}$. **Fine-tune Dataset** Given a set of small sample data for model training (fine-tuning data), denoted as $\mathbb{T}' = \{\boldsymbol{x}'_1, \ldots \boldsymbol{x}'_m, \ldots, \boldsymbol{x}'_q\}$, contains $m$ labeled samples and $q - m$ unlabeled samples, the label is $l_{\mathbb{T}'} = \{1, \ldots, c\}$. The fine-tuning data contains $C_{\mathbb{T}'}$ channels, and each channel contains $|\boldsymbol{x}'_i|$ data points. Data from multiple channels need to be analyzed together, we focus only on data from a single channel. Additionally, the data can be compared at any length, requiring only scaling, as described in the appendix A.4.

**Problem Definition** Pre-training data set $\mathbb{T}$, which does not contain labels, is used for contrastive learning of $p$ samples. The weight of the pre-trained model $M_{pre} = f(\boldsymbol{x}_i)$ is obtained. Then a small number of samples labeled with $m$ are used to fine-tune $M_{pre}$ to obtain the model $M_{tune} = f(\boldsymbol{x}'_i)$.

## 4 METHODOLOGY

### 4.1 DATA PLATFORM FILTERING.

Data platform filtering is a method of data pre-processing, but not all data is suitable for this method (as explained in Appendix A.5). For example, if we have a time series $\boldsymbol{x}_i$, which contains many platform-like parts, such as an electrocardiogram, we cannot simply remove the platform parts because they may be useful.

---

**Algorithm 1:** Data Platform Filtering(***DPF***)

---

**Input:** $\mathbb{T} = \{\boldsymbol{x}_1, \ldots, \boldsymbol{x}_p\}, \mathbb{T}' = \{\boldsymbol{x}'_1, \ldots, \boldsymbol{x}'_q\}, winscale$
**Output:** $\mathbb{T}_{step}, \mathbb{T}'_{step}$
1   $k$=A null array, $\mathbb{T}_{step}, \mathbb{T}'_{step}$=A null array;
2   $winsize = ceil(max(\boldsymbol{x}_i) - min(X_i)/2)$ or $ceil(max(\boldsymbol{x}'_i) - min(\boldsymbol{x}'_i)/2)$;
3   **for** $i = 1 : winsize : len(\boldsymbol{x}_i \; or \; \boldsymbol{x}'_i) - winsize$ **do**
4      $windata = \boldsymbol{x}_i(i, i + winsize)$ or $\boldsymbol{x}'_i(i, i + winsize)$;
5      $winTHR = abs(max(windata) - min(windata))$;
6      **if** $winTHR < winscale$ **then**
7         $k.add(i)$;

8   **for** $j=len(k):-1:2$ **do**
9      $\boldsymbol{x}_i = (\boldsymbol{x}_i(1 : k(j)), \boldsymbol{x}_i(k(j) + winsize : len(\boldsymbol{x}_i))$ or $(\boldsymbol{x}'_i(1 : k(j)), \boldsymbol{x}'_i(k(j) + winsize : len(\boldsymbol{x}'_i))$;
10      $\mathbb{T}_{step} = \mathbb{T}_{step} \cup \boldsymbol{x}_i or \mathbb{T}'_{step} = \mathbb{T}'_{step} \cup \boldsymbol{x}'_i$;
11   **Output** $\mathbb{T}_{step}, \mathbb{T}'_{step}$;

---

Here we perform platform filtering on $\boldsymbol{x}_i \in \mathbb{T}$ and $\boldsymbol{x}'_i \in \mathbb{T}'$, as shown in the figure 5. First, the algorithm calculates the maximum and minimum amplitudes in the data, and sets the sliding window size to half of the range. Based on the difference between the maximum and minimum amplitudes within the window, the algorithm determines whether to filter the current window. The advantage of this method is that it can remove the invalid parts of the data, and only retain the most prominent features in the time series.

## 4.2 SELF-ADAPTIVE FIR FILTER

We use a FIR filter (Wang et al., 2019) with a low-pass frequency designed to be adaptive. The value is set based on the maximum frequency of the current curve multiplied by $\sqrt{2}/2$. Since the parameters of the filter have a significant impact on the curve, an adaptive approach is used to avoid significant loss of detailed features, although it may not ensure optimal filtering. The analysis that we use FIR is shown in Appendix A.3. We will filter each sample $\boldsymbol{x}_i$ ($\boldsymbol{x}'_i$) based on the ratio of its maximum frequency in the dataset, and we call this the Self-Adaptive FIR Filter (**SAFF**) algorithm. Due to space limitations, we will omit the pseudocode for the algorithm.

---

**Algorithm 2:** Dataset Pre-process(***DP***)

---

**Input:** $\mathbb{T}, \mathbb{T}', k_{len}$
**Output:** $\mathbb{T}_{Filter}, \mathbb{T}'_{Filter}$
1   $\mathbb{T}_{Filter}, \mathbb{T}'_{Filter}$ = a null array, $freq_{max} = 0$;
2   **for** $i = 1 : len(\mathbb{T}) \; or \; len(\mathbb{T}')$ **do**
3      Get $\boldsymbol{x}_i$ or $\boldsymbol{x}'_i$'s frequency $\boldsymbol{x}_i^f$ or $\boldsymbol{x}'^f_i$.;

4   Obtain the max frequency $freq_{max}$;

5   $CutoffFrequency = \frac{\sqrt{2}}{2} \times freq_{max}$;
6   **if** $CutoffFrequency < 50$ **then**
7      Return;;

8   $winscale_{all}$ = statistic the $winscale$ value of each window.;
9   $winscale = IQR_1(sort(winscale_{all}))$;
10   $\mathbb{T}_{step}, \mathbb{T}'_{step}$ = **DPF**($\mathbb{T}$ or $\mathbb{T}'$, $winscale$);
11   **for** $i = 1 : len(\mathbb{T}_{step} \; or \; \mathbb{T}'_{step})$ **do**
12      **if** $len(temp) < k_{len}$ **then**
13         continue;
14      $temp_{fil}$ = **SAFF**($temp, order, Fs, CutoffFrequency$);
15      $\mathbb{T}_{Filter}.add(temp_{fil})$ or $\mathbb{T}'_{Filter}.add(temp_{fil})$;
16   **Output** $\mathbb{T}_{Filter}, \mathbb{T}'_{Filter}$;

---

## 4.3 DATA PRE-PROCESSING

Here we pre-process each sample in the dataset to improve the stability of the entire dataset. We use two methods including but not limited to platform filtering and filtering to achieve this. If the quality of the data itself is high, we need to make a preliminary judgment on the dataset. During the data platform filtering process, we need to make a reasonable division of the $winscale$ value, otherwise overfiltering may occur, which will lead to the loss of the dataset's features. As mentioned

in the FIR filtering process, data with frequencies lower than $\sqrt{2}/2$ of the maximum frequency are not filtered (see Appendix A.3). The pseudocode is shown in Algorithm 2.

## 4.4 MODEL STRUCTURE

Our work follows the Zhang et al. (2022)'s work, We adopted his concept of time-frequency consistency, but made some modifications on the model. We embedded a CNN module behind the transformer. The structure of the model is shown in figure 2.

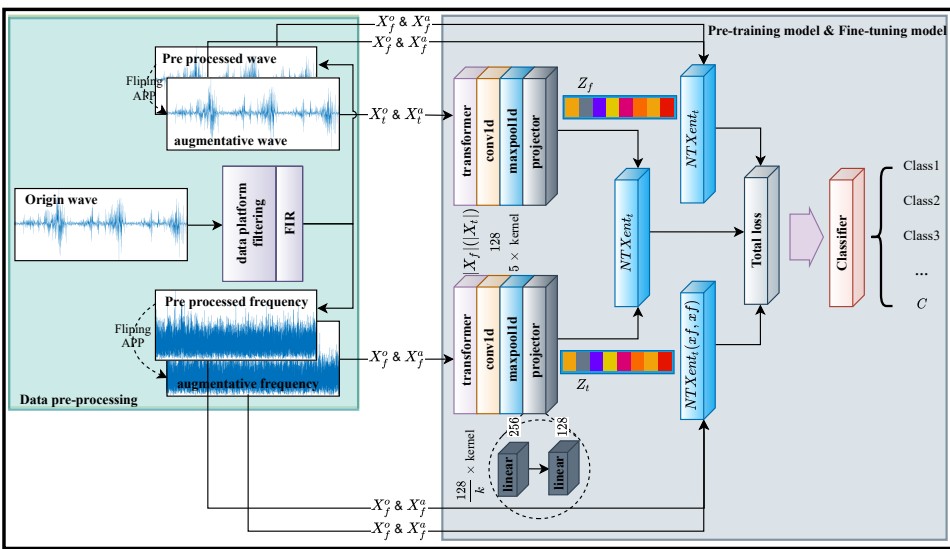

Figure 2: The Data Pre-processing Time Series Classification (DPTSC) model.

First, after the original waveform is filtered through the data platform and filter, data augmentation (see 4.5) is performed. The data is then divided into two parts are frequency domain and time domain, and then fed into the classification model for representation.

Second, the transformer continues to use the work of Zhang et al. (2022), using two transformers, one for the frequency domain and the other for the time domain, as two input encoders. Then, a CNN is used to convolve the data after the encoder to obtain a higher-dimensional feature vector. Here, CNN weakens the impact of peaks, which is very effective for data with abrupt changes in some amplitudes (such as BD). The transformer precisely increases the weight of the peak part of the data, while the CNN cuts off the high-dimensional features of the data to prevent the extraction of too many peak part features. This has been verified in previous work (Li et al., 2021). Afterwards, the model calculates the loss in the time domain and frequency domain, as well as their respective losses. After summing them up, the total loss is obtained and the classifier is trained.

## 4.5 SORTING SIMILARITY

NT-Xent loss function is widely used in contrast learning (Chen et al., 2020), (Tang et al., 2020), but in many cases, cosine similarity is essentially to measure the morphological difference between waveforms. When the morphological difference of time series itself is large, cosine similarity can no longer satisfy the fine similarity calculation. So we will modify the $sim(\boldsymbol{x}_i^{\mathbb{T}}, \boldsymbol{x}_j^{\mathbb{T}})$ part to cause NT-Xent loss. Our method presents a similar approach to the waveform re-construction work described in Zhang et al. (2023), but with the added step of sorting the waveforms by amplitude and using the new arrangement along the $x$ axis to determine the similarity between $\boldsymbol{x}_i$ and $\boldsymbol{x}_j$. The formula for this similarity is given by equation 1 and 2.

$$s(\boldsymbol{x}_i) = sort(\boldsymbol{x}_i), \boldsymbol{x}_i \in \mathbb{T}, \quad s(\boldsymbol{x}_j) = sort(\boldsymbol{x}_j), \boldsymbol{x}_j \in \mathbb{T} \tag{1}$$

$$sim(s(\boldsymbol{x}_i), s(\boldsymbol{x}_j)) = \frac{D(s(\boldsymbol{x}_i), s(\boldsymbol{x}_j))}{2} + \frac{D(|arg_t(s(\boldsymbol{x}_i)) - arg_t(\boldsymbol{x}_i)|, |arg_t(s(\boldsymbol{x}_j)) - arg_t(\boldsymbol{x}_j))|)}{2} \tag{2}$$

The $D(\cdot)$ present the Hausdorff distance function, and $arg$ denotes the $x$ axis values of $\boldsymbol{x}_i$, $s(\cdot)$ denotes sort the $\boldsymbol{x}_i$ according to amplitude. By sorting the $\boldsymbol{x}_i$ values and their corresponding $x$ axis coordinates and calculating the similarity between the resulting orderings, we can perform distance calculations between two sets of waveforms while avoiding inconsistencies in waveform shape that could lead to large within-class distances and interfere with model learning during fine-tuning.

For a similarity calculation based on NT-Xent loss, we replace the similarity function with the rank distance instead of the DTW distance, cosine similarity, or other computationally intensive distance functions. This is because many time series are very similar, and using the DTW distance is extremely time-consuming, while cosine similarity cannot cover the amplitude distance of waveforms. Inspired by the EMD method (Boudraa & Cexus, 2007), we sort the time series and calculate the Hausdorff distance between two time series, which preserves the time attribute while retaining the graphic features of the time series.

**Our model's advantage**

We use different similarity functions in the pre-training and fine-tuning stages. In the pre-training stage, we use cosine similarity because it is more lenient and aligns with the overall approach of pre-training. Even if the shapes of curves within a class are inconsistent, the weights are not fixed within a specific range, which helps prevent weight overfitting. In the fine-tuning stage, we use ranking similarity because it allows differentiation in waveform details, enabling the trained weights to achieve higher accuracy.

We process time series as a whole requires a significant amount of resources, determine the parameters is crucial. For data platform filtering, an adaptive strategy is needed for $winscale$. A simple approach is to calculate the $winscale$ value for each window and then derive the $winscale$ sequence statistics. The $Q_1/2$ is then taken as the final platform filtering value. This corresponds to a lower threshold, which helps minimize the platform filtering threshold as much as possible.

In contrast learning, positive and negative samples are used. We use APP and Flipping augmentation (Wen et al., 2020) to generate positive samples. In each batch, all samples except for the augmented ones are considered negative. Each sample can be transformed into three positive samples. If more positive samples are needed, additional augmentation techniques can be used.

## 5 EXPERIMENTS

### 5.1 EXPERIMENTS SETUP

We are using the Windows platform with an Intel Core i9-11900H 2.5GHz CPU, an RTX 3060 graphics card with 11GB of memory, a 1TB hard drive, and 16GB of RAM. We use the Python and MATLAB programming languages to compare 5 baseline algorithms. We pre-train on four pairs of datasets and evaluate the model performance using 5 metrics: Accuracy, Precision, F1-score, AUROC, and AUPRC. All experimental results were obtained by running three times and taking the average value.

### 5.2 DATASET

**SleepEEG**. The sleep-edf database has been expanded to contain 197 whole-night PolySomno-Graphic sleep recordings. The EOG and EEG signals were each sampled at 100Hz. The submental-EMG signal was electronically highpass filtered, rectified and low-pass filtered after which the resulting EMG envelope expressed in uV rms (root-mean-square) was sampled at 1Hz. The data has been divided into 5 classes and can be obtained from https://www.physionet.org/content/sleep-edfx/1.0.0/. **Epilesy**. Indications of nonlinear deterministic and finite-dimensional structures in time series of brain electrical activity, which are divided into 2 classes: Dependence on recording region and brain state. The data can be obtained from https://repositori.upf.edu/handle/10230/42894. **BD-A** and **BD-B**. The BD was collected from vibration experiments conducted under 4 different condi-

tions, with a sampling rate of 64KHz. The data has been divided into 3 classes for its large scale and divivded in 2 groups BD-A and BD-B, respectively, which can be obtained from https://mb.uni-paderborn.de/en/kat/main-research/datacenter/bearing-datacenter/data-sets-and-download.  **HAR**. Body posture data, including 6 classes, sampling rate is 50Hz, data are available on https://archive.ics.uci.edu/ml/datasets/Human+ Activity+Recognition+Using+Smartphones.  **Gesture**. Gestures recognition data, including 8 classes, 100Hz sampling rate, data can be obtained from http://www.timeseriesclassification.com/description.php?Dataset=UWaveGestureLibrary. **Microquake**. Microquake data, including 3 classes, sampling rate of 5KHz.  **Earthquake**. Seismic data, including 4 classes, sampling rate 10Hz or 12Hz and etc., data can be obtained from https://ds.iris.edu/mda/?type=assembled, we download 10 dataset from it and they have 1156 samples. Details of the datasets are shown in Table 1, the table shows that we used fine-tuning on a small sample set, and the number of pre-training samples was not large-scale.

| Situation | Kind | Dataname | Samples | Channels | Classes | Length | Freq.(Hz) |
|---|---|---|---|---|---|---|---|
| SleepEEG→Epilepsy | Pre-training | SleepEEG | $198,032$ | 1 | 5 | 3000 | 100Hz |
| | Fine-tuning | Epilepsy | $1,500$ | 1 | 2 | 178 | 178 |
| BD-A → BD-B | Pre-training | BD-A | $8,184$ | 1 | 3 | 2400 | 64K |
| | Fine-tuning | BD-B | 400 | 1 | 3 | 2400 | 64K |
| HAR → Gesture | Pre-training | HAR | $10,299$ | 3 | 6 | 512 | 50 |
| | Fine-tuning | Gesture | 200 | 1 | 8 | 315 | 100 |
| Earthquake → Microquake | Pre-training | Earthquake | 1156 | 6 | 4 | 10K | 5K |
| | Fine-tuning | Microquake | 153 | 6 | 3 | 10K | 5K |

Table 1: Dataset information.

**Baseline.**

**LB-simTSC**(Xi et al., 2023) This method uses DTW for distance similarity calculation on curves, and then utilizes GCN for supervised classification on time series. This method calculates pairwise similarity between all $K$ samples, so the size of the graph depends entirely on the value of parameter $k$. The determination of $k$ determines the accuracy of the final data. According to the context of the paper, $k = 10$. **DL-HMM**(Ghimatgar et al., 2020)This method involves three steps: feature selection, channel selection and BiLSTM training, and HMM correction. Since we are using a single channel, we have omitted the channel selection step here. **KNN**(Keller et al., 1985) Using an unsupervised method, we are using the KNN method under the sklearn package. **SSTSC**(Xi et al., 2022) It is a semi-supervised model that provides supplementary weights for temporal context information. Since it uses a small amount of labeled data, we take $20\%$ of the data as the labeled data set. **TF-C**(Zhang et al., 2022) It is a self-supervised pre-training fine-tuning model that does not change its parameters.

### 5.3 EXPERIMENTS ANALYSIS

**Fine-tuning Analysis**.

We compared our model with 5 baseline algorithms after fine-tuning with 4 datasets. The results are shown in Table 2, where the header represents the metrics, the first column represents the algorithm names, the second column represents the pre-training and fine-tuning scenarios, and the remaining columns represent the experimental results.

The table shows that DPTSC performs well on all datasets. LB-SimTSC uses waveform similarity for classification, which works well when the data within each class has different shapes, but in reality, data cannot always be distinguished by shape. SSTSC expands the sample size by splitting and combining samples, but this method divides each sample into three parts and swaps them randomly, which can cause conflict in the weights used during supervised training and affect classification accuracy. The addition of unlabeled data may also interfere with classification accuracy. The TF-C model lacks data pre-processing, which leads to lower accuracy. The KNN model is unsupervised, so its classification accuracy depends on the similarity of shapes within each class, rather than other factors. Therefore, it works better when the shapes within each class are similar but the shapes between classes are different, such as in the SleepEEG and Epilepsy training scenarios. DL-HMM uses a hidden Markov model for post-processing of classification results, but training an HMM is not the same process as training a BiLSTM model, which can lead to discrepancies between the two methods. Therefore, using HMM as a benchmark is appropriate.

| Models | Situations | Acc | Precision | Recall | F1 | AUROC | AUPRC |
|---|---|---|---|---|---|---|---|
| DPTSC(Ours) | Sleep→Epilesy | 96.01 | 97.01 | 97.00 | 99.10 | 99.08 | 99.19 |
| | BD-A→BD-B | 94.35 | 94.60 | 94.35 | 94.38 | 98.47 | 97.21 |
| | HAR→Gesture | 84.05 | 85.06 | 89.78 | 87.60 | 89.70 | 89.50 |
| | Earthquake→Microquake | 88.79 | 87.06 | 89.18 | 86.99 | 89.61 | 90.76 |
| KNN | Sleep→Epilesy | 85.25 | 86.39 | 64.31 | 67.91 | 64.34 | 62.79 |
| | BD-A→BD-B | 44.73 | 28.46 | 32.75 | 22.84 | 49.46 | 33.07 |
| | HAR→Gesture | 67.66 | 65.00 | 68.21 | 64.42 | 81.90 | 52.31 |
| | Earthquake→Microquake | 62.79 | 65.27 | 63.17 | 65.22 | 66.78 | 66.52 |
| TF-C | Sleep→Epilesy | 94.95 | 94.56 | 89.08 | 91.49 | 98.11 | 97.03 |
| | BD-A→BD-B | 89.34 | 92.09 | 85.37 | 91.62 | 94.35 | 95.27 |
| | HAR→Gesture | 78.24 | 79.82 | 80.11 | 79.91 | 90.52 | 78.61 |
| | Earthquake→Microquake | 80.01 | 82.71 | 83.44 | 85.09 | 86.78 | 86.52 |
| DL-HMM | Sleep→Epilesy | 90.74 | 92.39 | 91.77 | 93.21 | 95.34 | 92.19 |
| | BD-A→BD-B | 55.34 | 59.22 | 78.06 | 65.34 | 72.01 | 67.99 |
| | HAR→Gesture | 61.29 | 58.18 | 59.87 | 62.31 | 65.98 | 71.39 |
| | Earthquake→Microquake | 80.72 | 80.01 | 83.23 | 79.04 | 81.66 | 85.79 |
| LB-SimTSC | Sleep→Epilesy | 67.13 | 69.06 | 64.88 | 72.87 | 69.35 | 74.56 |
| | BD-A→BD-B | 59.68 | 57.39 | 60.79 | 57.12 | 55.59 | 60.12 |
| | HAR→Gesture | 56.29 | 49.36 | 51.71 | 56.73 | 54.00 | 57.09 |
| | Earthquake→Microquake | 65.10 | 62.39 | 67.01 | 69.91 | 72.22 | 65.09 |
| SSTSC | Sleep→Epilesy | 74.16 | 71.21 | 79.66 | 73.38 | 78.21 | 77.31 |
| | BD-A→BD-B | 69.15 | 64.19 | 65.21 | 68.81 | 61.45 | 69.34 |
| | HAR→Gesture | 54.69 | 59.44 | 59.36 | 61.81 | 54.11 | 58.18 |
| | Earthquake→Microquake | 62.20 | 65.35 | 63.25 | 67.59 | 71.28 | 72.35 |

Table 2: Different indicators of different datasets.

**Sorted Similarity Representation**.

This section aims to verify the effectiveness of sort similarity by representing the data arrangement calculated using sort similarity, as shown in figure 3. The $x$ and $y$ axes represent the relative positions formed by the two curves through two-dimensional coordinates.

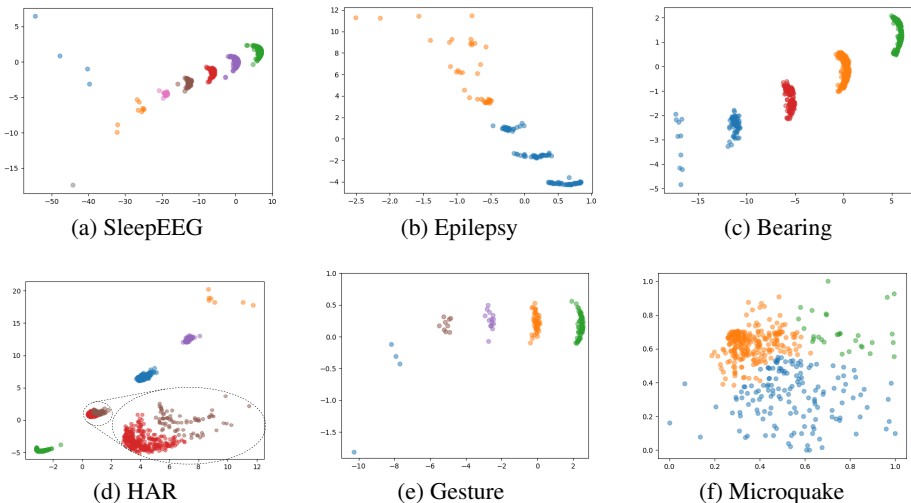

Figure 3: Sorted loss representation.

The image shows that samples with the same label are placed closer to each other, while samples with different labels are placed farther apart. This indicates that the sort similarity is effective. When the HAR data representation graph is zoomed in on local positions, the data differentiation remains clear. The Gesture representation graph shows that the data is divided into 4 classes, although the text mentions that Gesture is divided into 2 classes. The Microquake data have a high degree of

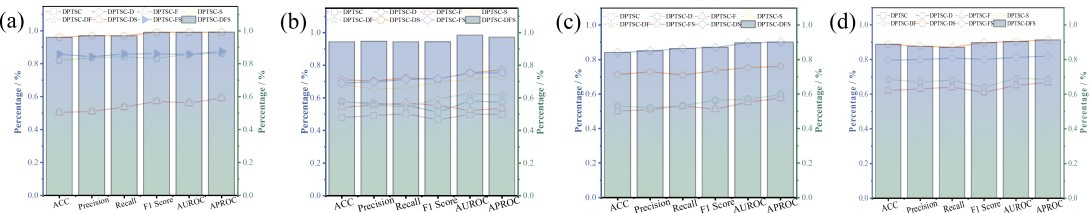

Figure 4: Ablation study.

sorting similarity, so the distribution of coordinates within each class is not concentrated during representation, but the distance between different classes is still clear when viewed as a whole. Therefore, the representations of all datasets effectively separate data from different classes, which is advantageous for the training phase of fine-tuning.

**Ablation study**

We compared 7 combinations of DPTSC, DPTSC-D, DPTSC-F, DPTSC-S, DPTSC-DF, DPTSC-DS, and DPTSC-FS, and DPTSC-DFS by using data platform filtering as condition D, filtering as condition F, and sort similarity as condition S. We conducted experiments on each method and observed the changes in 6 indicators to analyze the impact of different modules on model performance.

From the figure, it can be seen that the performance of the original DPTSC model is the lowest. The performance of DPTSC-S is only second to that of DPTSC-DFS. From the results of the BD, the filtering module F itself does not significantly improve the performance because the data itself contains too many invalid parts (See Appendix A.3). At the same time, both pre-training and fine-tuning stages use cosine similarity, which amplifies the differences in the shape of the data. The phase of the BD itself has an offset, which makes it difficult for the cosine similarity calculation to converge, resulting in poor performance. Using only ranking similarity for training, the invalid parts and high-frequency parts in the data will also interfere with the final results of the model, making the overall performance lag behind. Therefore, the D, F, and S modules must be used simultaneously to achieve optimal performance.

## 6 DISCUSSION

We improves the accuracy of the model by using data pre-processing to filter out interference, using a CNN module to weaken the impact of mutation, and finally using sorting similarity to replace cosine similarity. Through fine-tuning on 8 datasets, our algorithm performs better than other algorithms. From the accuracy, it can be seen that pre-training can achieve training without labeled data, greatly reducing the cost of labeling data, which is of great significance for practical applications.

Future work will mainly focus on the intelligence of the pre-processing parameters of the model itself, because the interference items in the data itself are not the same, so the depth and breadth of data filtering and platform processing have a great impact on the results. In addition, the loss similarity of the model itself also needs to be re-constructed based on the characteristics of time series, rather than relying solely on sorting.

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

## A APPENDIX

### A.1 PARAMTERS IN PRE-PROCESS AND MODEL

In this part of the analysis model, we used fixed values for the data platform filtering window thresholds on 8 datasets. These values, in the order of the datasets, are 0.05, 0.05, 0.1, 0.2, 0.2, 0.1, 0.1, respectively. Additionally, the maximum frequencies for FIT filtering are 7.04Hz, 13.38Hz, 4887.62Hz, 6203.29Hz, 0.57Hz, 2.78Hz, 20.15Hz, 12.78Hz, respectively. The training lengths of 4 datasets are 178, 2400, 206, and 1500, respectively. The training iteration numbers are 80, 100, 100, and 100, respectively. The temperature coefficient is set to 0.2, the batch size during the pre-training phase is 128, and the batch size during the fine-tuning phase is 42.

In the experiments, we found that the best performance for the CNN was achieved when the kernel size was 5 times the size of the pooling layer. The maximum pooling layer size is $5 \times kernel$.

### A.2 DATA PLATFORM FILTERING

Data platform's part of consequences is shown as figure 5.

From the figure, it can be seen that the platform filtering has no effect on the SleepEEG and epilepsy data because there are no relatively flat parts in these two datasets. However, there are some flat parts filtered out by the platform for other datasets. From the figure 5 (c)-(f), it can be seen that the bearing, the Gesture and HAR, and the Microquake have been filtered some parts of them, among which the Gesture and Microquake have been filtered more significantly. The reason is that there is a waiting time before the start of the Gesture data, and the Microquake data is truncated to include the part with a smaller energy after the vibration part of the data. It can be seen that the platform filtering strategy we proposed has played a corresponding role in filtering out invalid parts of the data.

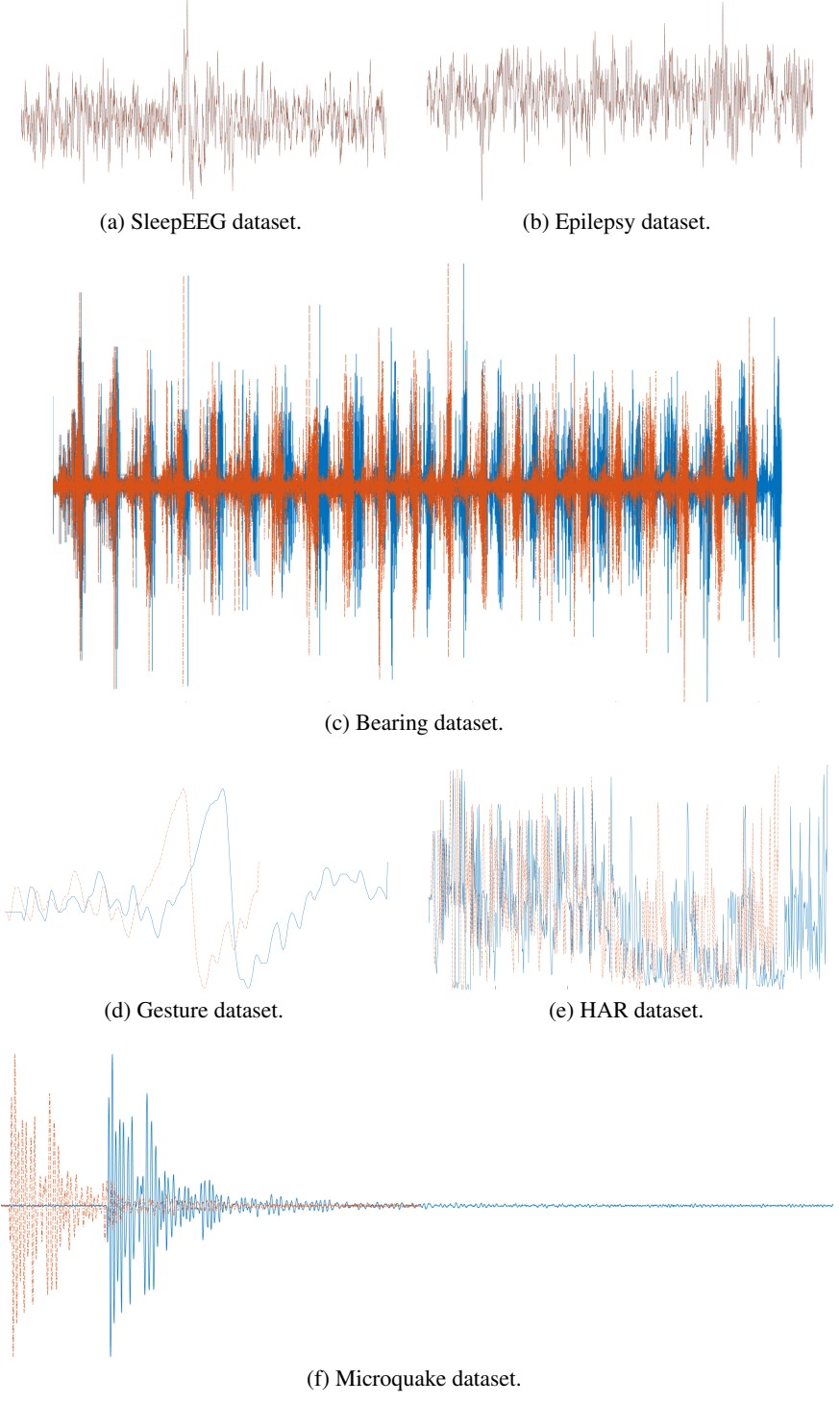

(a) SleepEEG dataset.

(b) Epilepsy dataset.

(c) Bearing dataset.

(d) Gesture dataset.

(e) HAR dataset.

(f) Microquake dataset.

Figure 5: The Epilepsy, Gesture and HAR datasets after data paltform filtering.

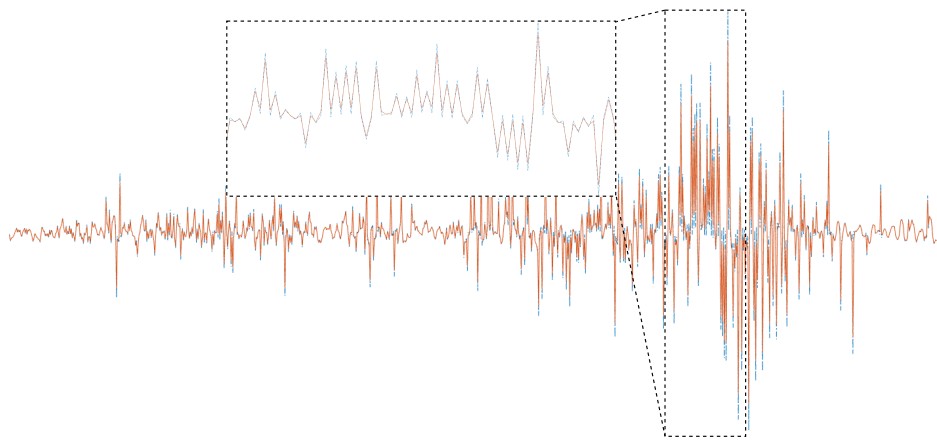

Figure 6: FIR filter for BD dataset.

## A.3 FIR FILTERING

A FIR filter can be denoted as equation 3 and 4.

$$y(n) = \sum_{k=0}^{N-1} g(k)x(n-k) \tag{3}$$

$$G(z) = \sum_{k=0}^{N-1} g(k)z^{-k} = g(0) + g(1)z^{-1} + \cdots + g(N-1)z^{-(N-1)} \tag{4}$$

In equation 3, $h(k)$ represents the coefficients of the filter and $x(n-k)$ represents $x(n)$ delayed by $k$ cycles. The transfer function of the system $H(z)$ can be expressed as equation 4. From equation 4, it can be seen that the filtering process is mainly a convolution process completed by a specific set of coefficient pre-signals. From equation 4, it can be seen that there are $N-1$ zeros in a limited $z$-plane, and all $N-1$ poles are located at $z = 0$. Therefore, the FIR filter belongs to a full-zero-point filter, which is a stable system with a finite-length unit impulse response. When the coefficients of the FIR filter meet certain conditions, its phase-frequency characteristics are linear, which can effectively preserve the phase information of the signal. Therefore, FIR filters have a wide range of applications.

We used FIR low-pass filters at different frequencies for each dataset, and calculated the cutoff frequency at the $\sqrt{2}/2$ position based on the maximum frequency of each dataset. The results of the data processing are shown in the figure 6, and the cutoff frequencies corresponding to the maximum frequencies of each dataset are shown in the table 3.

| Dataset | SleepEEG | Epilepsy | BD-A | BD-B | HAR | Gesture | Earthquake | Microquake |
|---|---|---|---|---|---|---|---|---|
| Cut-off frequency(Hz) | 7.04 | 13.38 | 6203.29 | 4887.62 | 0.57 | 2.78 | 0.01 | 0.2 |

Table 3: Cut-off frequency Datasets

As the high-frequency components of other data do not exceed 50Hz, we only show the filtering results of the bearing dataset. From the figure, it can be seen that the high-frequency components of the BD itself can be reduced in peak value and noise can be eliminated by using a low-pass filter.

## A.4 UNEQUAL LENGTH DATA PROCESSING

Since most time series data are of unequal lengths, it is sufficient to maintain the shape of the time series in both the frequency and time domains without significant changes. Therefore, traditional interpolation/sampling methods can be used. If the training data is longer than the prediction data

and exhibits unstable cycles, a stable portion can be extracted. If the training data is shorter than the prediction data, interpolation methods can be employed.

## A.5 DISCUSSION OF DATA PRE-PROCESSING

Due to the presence of noise in real-world data and various sensor-related issues such as data offset and drift, pre-processing is essential to address the interfering frequency signals in the data (Wang et al., 2019). For certain types of data such as electrocardiogram data, there may be platform components that carry valuable information for disease diagnosis (PETERS et al., 1992).

