# OpenReview forum: "A Self-Supervised Pre-Training Model for Time Series Classification based on Data Pre-Processing"
_ICLR.cc/2024/Conference — Submitted to ICLR 2024_

### Official Review · Reviewer_GpZ4 · 2023-10-30

**Soundness:** 1 poor
**Presentation:** 1 poor
**Contribution:** 1 poor
**Rating:** 1
**Confidence:** 4

**Summary:**

This paper deals with the task of pre-training time series classifiers.
It relies on (Zhang et al. 2022) and modifies this approach in two ways: (i) a pre-processing step is proposed to filter out uninformative parts from the time series and (ii) the similarity computation used in the NT-Xent loss is replaced by a similarity measure that relies on sorted version of the time series.

**Strengths:**

The problem that is tackled is an important one.

**Weaknesses:**

I can see two main weaknesses for this paper:

1. The presentation of the paper is so unclear that one cannot fully grasp the main take-away message from the paper.

* Introduction is close to non-informative
* In the Related Work section, competitors are described at such a high level that one cannot understand their limitations
* Even the notations in Section 3 are confusing
    * The text discusses multivariate time series dataset (each dataset comes with a number of channels) and then, suddenly: "Data from multiple channels need to be analyzed together, we focus only on data from a single channel", yet, in the experiments, some datasets are multivariate...
    * Problem definition is unclear, since the same function $f$ is used to map a pre-training sample to the pre-trained model $M_{pre}$ and a "fine-tune sample" (with the same index...) to the final model $M_{tune}$
* When it comes to the specific novelty of the paper:
    * Algorithms 1 and 2 lack rigor (ex: in Algo1, is $i$ the index of a time series (as it seems to be in $x_i$) or a time instant (as it seems to be in the slicing $i:i+winsize$)?)
    * NT-Xent description is not clear enough. Authors write "inspired by the EMD method (Boudraa & Cexus, 2007), we sort the time series and calculate the Hausdorff distance between two time series, which preserves the time attribute while retaining the graphic features of the time series.": I did not know this paper, had a look at it and it seems this (Boudraa & Cexus, 2007) paper does not mention the idea of sorting the time series nor using the Hausdorff distance between them (it consists in decomposing the signal).

2. The novelty is quite unclear, since the paper relies on a well-established method (Zhang et al. 2022) and slightly modifies it by adding a pre-processing step and changing the similarity measure in the loss.
Since these two points constitute the novelty of the paper, they should be motivated, discussed and illustrated in many more details than what is done in the current version of the paper.

**Questions:**

Given my comments above, I do not think answering these questions could be sufficient to change my mind.

* Could you elaborate on the similarity measure that is introduced in Section 4.5, and more specifically on the links of this with the EMD method that is said to be a source of inspiration?
* What representation is used in Figure 3? Is it tSNE? MultiDimensional Scaling? Something else?
* Why do you evaluate on only 4 pairs of datasets while timeseriesclassification.com hosts many more datasets, arranged in categories?

---

> ### Author Response · Authors · 2023-11-13
> **Reply to GpZ4**
>
> First of all, thank you very much for reviewing our work and providing valuable suggestions, as well as recognizing some of our work. We will continue to work hard on my research. Here is my response to the questions you raised.
>
> Q1:
> The similarity measurement is based on EMD for two main reasons. Firstly, in our actual projects, we use EMD decomposition to calculate the energy of waveforms. However, the decomposed waveforms are relatively smooth and lose some details, while real waveforms are generally complex in shape. Therefore, we thought of sorting time series and directly comparing their similarity. Secondly, considering that similar waveforms within the same category have higher similarity after sorting, we use the Hausdorff distance, which calculates the maximum distance between two waveforms. Sorting is equivalent to organizing the waveforms in advance, and the loss function in the fine-tuning stage requires refinement.
>
> Q2:
> We use the fit_transform function in the Sklearn library to convert the high-dimensional features of each waveform into a two-dimensional coordinate system, and plot the resulting graph, which is similar to the t-SNE method you mentioned.
>
> Q3:
> We evaluated four sets of data from representative fields, and of course, other data can also be evaluated. For example, traffic data has a lower sampling rate and cannot reflect the advantages of our method in this paper. Therefore, we selected data with a higher sampling rate for evaluation.
>
> Regarding the issues you raised about the expression of the article, I sincerely accept the criticism because my English is not good.
>
> Firstly, we humbly accept suggestions regarding the novelty of the article. However, the focus of the method proposed in this paper is on the data filtering part. Most existing articles mainly evaluate the generalization ability of models, while neglecting the weaknesses of the models themselves. If we only consider improving the models themselves, all current works have done very well. However, the field of time series analysis lacks an excellent data preprocessing method, which should be effective for the majority of data.
>
> Secondly, you mentioned that the symbol $x_i$ is ambiguous in section 3 of the article. Here, $i$ represents the sample index. It is indeed a mistake on our part. In line 2 of Algorithm 1, it should be $winsize=ceil(max(x_i)-min(x_i))/2$ or $ceil(max(x'_i)-min(x'_i))/2$.
>
> Thirdly, the ablation study compares the accuracy between using FIR filters and not using FIR filters. It is denoted as F. The use of sorting similarity is denoted as S, and the use of platform filtering algorithm is denoted as D. The original text is located in the first paragraph under "Ablation Study" as follows.
>
> “We compared 7 combinations of DPTSC, DPTSC-D, DPTSC-F, DPTSC-S, DPTSC-DF, DPTSCDS, and DPTSC-FS, and DPTSC-DFS by using data platform filtering as condition D, filtering as condition F, and sort similarity as condition S.”
>
> Lastly, regarding the issue of multivariate channels, we did mention in the article that it is applicable to multivariate sequences. However, we did not specifically mention methods for fusing multiple channels. We can use methods such as $\sqrt{x_{1i}^2+x_{2i}^2+\cdots+x_{ni}^2}$ to fuse data from multiple channels, etc.

---

### Official Review · Reviewer_QovG · 2023-10-31

**Soundness:** 2 fair
**Presentation:** 1 poor
**Contribution:** 1 poor
**Rating:** 3
**Confidence:** 3

**Summary:**

The authors describe modifications to an existing pre-training and fine-tuning time-series classification model to filter extremities in the data during pre-processing, and utilizing an ordered similarity measures, to show improvements in overall classification metrics across various domain transfer datasets. Particularly, the improved model pre-processes the data using adaptive low-pass FIR filter based on maximum and minimum amplitudes in the window of computation. This data is then divided to obtain frequency and time domain embeddings using separate transformer architectures as per existing work. However, the authors slightly modify the architecture by adding a waveform sorting function over amplitudes in the loss function where Hausdorff distance function is used instead of cosine similarity (employed only at the fine-tuning stage and not pre-training stage). They empirically demonstrate that these two modifications enhances the overall accuracy and F1 scores across various task transfers where model is pre-trained on one dataset using self-supervision, and fine-tuned on another dataset having a small amount of labels.

**Strengths:**

1. The empirical experiments shows good comparisons with similar base-line mechanisms, clearly illustrating the advantages of the modifications introduced by the authors to the existing model.
2. By depicting the sorted loss representation, the authors clearly illustrate key concepts to describe need for capturing morphology of class differences in datasets, among the datasets they have used.
3. The authors explain competitive approaches in detail, providing intuitive reasons for lower scores obtained by those models in their empirical study.

**Weaknesses:**

1. The paper is hard to follow for readers not in the domain of time-series classification. Particularly, the motivation in the introduction of the paper seems to start with applications in what the authors say as industrial field. But, it is very hard for a naive reader to quickly understand the context of the paper in the introduction. The context of the problem under consideration becomes clear only after reading some cited papers, particularly that of Zhang et al. (2022) which the authors extensively employ and modify. For example, in section 2.1, sentences such as "DTW cannot obtain the more information of curves ..." or "... due to issue of the rationality of segment combinations during classification" etc. is very confusing or imprecise. It is hard to understand what the authors intended to say here.
2. Notations used to mathematically describe the problem and algorithms are imprecise and confusing. For example, in section 3, it is unclear if CT = CT'? Is there any relationship between T and T'? Is Mpre = f(xi) a function of single data point within a channel or is xi a sample? In Algorithm 2, where is Tstep used? What is the temp variable?
3. In Figure 2, it would be better to highlight modified parts of Zhang et al's model. Particularly, it is unclear if CNN is part of the modification or the original model. If CNN is part of the modification, then it's applicability on the transfer learning empirical study is incomplete.
4. It is good that the authors use Hausdorff distance to preserve time attributes and retain graphical features. However, there are various other distance measures such as Bhattacharya distance, or EMD method, that could also be used instead. It would be good if the authors empirically show evidence why Hausdorff was chosen.
5. From Table 2, it is clear that DPTCS is better than the TF-C method that the authors have modified. However, the original Zhang et al., 2022 article evaluates one-to-many domain adaptation which brings out areas where other competing models perform better than TF-C. This key evaluation is missed by the authors in the current submission.

**Questions:**

Please see weaknesses section. In addition, I have a few additional clarifications here:

In section 4.1, algorithm 1, it is unclear how XI is updated. Can you please explain the reason and methodology behind line 9?

In section 4.1, is Figure 5 incorrectly specified as algorithm 1?

In algorithm 2, what is IQR?

In the introduction, what is NT-Xent loss?

---

> ### Author Response · Authors · 2023-11-13
> **Reply to QovG**
>
> First of all, thank you very much for reviewing our work and providing valuable suggestions, as well as recognizing some of our work. We will continue to work hard on my research. Here is my response to the questions you raised.
>
> Q1:
> In Algo1, lines 3-7 save all starting positions with window thresholds less than $winTHR$ to $k$. Lines 8-10 update the sample data by looping through $k$ and removing each platform that is less than $winTHR$. Here, we ignored the issue of looping through the entire sample set $\mathbb{T}$. A $for$ loop should be inserted before line 8 to select each sample one by one, and then perform the operations in lines 3-10 on each sample. The added line is as follows:
>
> $for\ m=1:len(\mathbb{T}):1$
>
> Q2:
> There is no error specification in Figure 5. The waveform in Figure 5 (a) and (b) does not have a platform part, so the waveform after processing is consistent with the original shape. The waveforms in (c), (d), (e), and (f) can demonstrate the effect of platform filtering.
>
> Q3:
> IQR (Interquartile range) refers to the concept of quartile range in statistics.
>
> Q4:
> NT-Xent loss is a commonly used loss function in contrastive learning (from A Simple Framework for Contrastive Learning of Visual Representations by Ting Chen et al.). The formula is as follows.
>
> $l_{i,j}=-log\frac{exp(sim(z_i,z_j)/\mathbb{T})}{\sum_{k=1}^{2N}[k\neq i]exp(sim(z_i,z_k)/\mathbb{T})}$
>
> Here, the batch contains N samples, and there are $2N$ samples because there are two branches. Except for the augmented image corresponding to itself, the other $2N-2$ should be regarded as negative pairs. In the formula, i and j are positive pairs, and the denominator is negative pairs.
>
> NT-Xent looks like a softmax function. This is because it adds vector similarity and temperature normalization factors. The similarity function is just cosine distance. Another difference is that the value in the denominator is the signed distance from positive examples to negative examples. It is not much different from CrossEntropyLoss. The intuition is that we want our similar vectors to be as close to 1 as possible because $-log(1) = 0$, which is the best loss. We want negative examples to be close to 0 because any non-zero value will reduce the value of similar vectors.
>
> Regarding the issues you raised about the expression and grammar in the article, I sincerely accept the criticism because my English is not good.
>
> Regarding the weaknesses you pointed out, here are our responses:
>
> 1. DTW itself is a method for measuring the similarity between two time series. The method proposed by Xi et al. (2023) does not consider the possibility that time series within the same class may have completely different shapes, while some time series between different classes may have similar shapes. This can lead to the method failing in scenarios with certain noise interference.
>
> 2. $C_\mathbb{T}$ and $C_\mathbb{T'}$ denote as pre-training data and fine-tuning data’s channels, respectively. $M_{pre}=f(x_i)$ denotes the input is pre-training dataset $\mathbb{T}$, and $M_{tune}=f(x'_i)$ denotes the input is pre-training dataset $\mathbb{T'}$, and $f(\cdot)$ denotes the mapping from each sample from $\mathbb{T'}$ to label $l_\mathbb{T'}$. Temp is a sample from $\mathbb{T}_{step}$ or $\mathbb{T'}_{step}$. And $\mathbb{T'}_{step}$ is used to self-adaptive filter.
>
> 3. We apologize for the incomplete explanation in the article. The CNN module was added based on the research by Zhang et al., and it was introduced to mitigate the impact of abrupt changes in the waveform. If we consider it incomplete, we should include ablation experiments to verify the impact of adding or not adding the CNN module on the experimental results.
>
> 4. Hausdorff distance is a method for calculating the maximum distance between two time series. Since this article uses the sorting similarity calculation method, using Hausdorff distance allows us to calculate the distance between time series with higher sorted similarity, which better reflects the distinguishing details between time series, rather than using cosine similarity or other distance calculation methods.
>
> 5. I believe this evaluation is not necessary because we compared the accuracy of TF-C with our model in the experiment, and we also conducted comparative experiments in the ablation study section. If we compare the accuracy from the ablation study with TF-C, it can also demonstrate the advantages of our method.

---

### Official Review · Reviewer_nrKg · 2023-10-31

**Soundness:** 1 poor
**Presentation:** 1 poor
**Contribution:** 1 poor
**Rating:** 3
**Confidence:** 4

**Summary:**

This paper presents a few modifications to an existing pre-training (contrastive learning)-based time series classification model (Zhang et al. 2022). Experimental results on 8 sets of time series show that the proposed modifications improve the classification accuracy.

**Strengths:**

1. The idea of adding a sorting-based loss seems to be interesting.

**Weaknesses:**

1. There are a few self-supervised contrastive learning-based models for time series classification as mentioned in Section 2.2. However, their limitations (and differences with the proposed work) have not been clearly discussed. There are several sentences at the end of Section 2.2 on this point but they are vague and difficult to follow. It is difficult the evaluate the novelty of the proposal.

2. The overall writing and use of language is substandard and requires substantial improvement. Many technical details are missing or inaccurate:

"Our work follows the Zhang et al. (2022)’s work, We adopted his concept of time-frequency consistency, but made some modifications on the model. We embedded a CNN module behind the transformer": What exactly are the modifications? Just adding a CNN module after the transformer?

"Given a set of time series that needs to be pre-trained": How do you pre-train a set of time series?

"Each sample consists of CT channels, and each channel contains |xi| data points:" What is a sample? A time series? What is a channel? What is a data point? And how are the |xi| data points of a channel formed?

Where does "temp" in Algorithm 2 come from?

3. The experimental results are not too convincing. The URC time series repository has over 100 sets of time series. It is unclear how the 8 sets are chosen for the experiments. Also, each chosen set requires a different hyperparameter setting as shown in Appendix A.1. It is unclear how the proposed model can be used in a real application.

4. Other presentation issues:
Statements like "The extensive use of time series in the industrial field (Bi et al., 2023)(Li et al., 2020)(Gupta et al., 2020) is beyond doubt." are too strong and may need to be tuned down.

"figure 1 shows the entire process" => "Figure 1 shows the entire process"

Incomplete sentence: "pre-training a classification model for sequential"

**Questions:**

See the weak points.

---

> ### Author Response · Authors · 2023-11-13
> **Reply to nrKg**
>
> First of all, thank you very much for reviewing our work and providing valuable suggestions, as well as recognizing some of our work. We will continue to work hard on my research. Here is my response to the questions you raised.
>
> Q1:
> The method proposed in this article focuses on the data filtering part. Most existing articles mainly evaluate the model's generalization ability and ignore the weaknesses of the model itself. If we only consider improving the model itself, all current work has been done very well. However, the field of time series lacks an excellent data preprocessing method, and this method should be effective for most data.
>
> Q2:
> Yes, we only added a 1convCNN module after the Transformer to weaken the abrupt changes in the time series. However, overall, the experimental results are still good.
>
> We modify to train a model as replace.
>
> $C_\mathbb{T}$ represents all channels of a multivariate time series, where each channel is a time series $x_i$. The length of a time series (sample) is finite, and the number of data points is $|x_i|$. $\mathbb{T}$ represents a pre-training dataset, and $\mathbb{T'}$ represents a fine-tuning dataset. $\mathbb{T}$ does not have labels, while $\mathbb{T'}$ contains a small number of labels for fine-tuning the model.
>
> In Algorithm 2, "temp" represents a sample extracted from $\mathbb{T}{step}$ and $\mathbb{T'}{step}$.
>
> Q3:
> We evaluated 4 sets of representative data in different fields, and other data can also be evaluated, such as traffic data, which has a low sampling rate and cannot reflect the advantages of the method in this paper. Therefore, we selected data with a high sampling rate for evaluation.
>
> Q4:
> Modified to: Time series are widely used in the industrial field.

---

### Official Review · Reviewer_PC6z · 2023-11-01

**Soundness:** 2 fair
**Presentation:** 1 poor
**Contribution:** 2 fair
**Rating:** 3
**Confidence:** 3

**Summary:**

This work proposes a pipeline for data pre-processing and training a classifier for time series. It pre-processes data with filters to retain the most prominent features of the data, along with some data augmentation to increase the amount of available pre-training data. Next, the data goes into the model for pre-training and fine-tuning. It first goes through two transformers, one for the frequency domain and the other for the time domain. Then, it is passed through CNNs to remove any high-dimensional features. Finally, the model calculates the loss in time and frequency domains. The sum of these losses is used to train the classifier. This work uses a modified version of NT-Xent loss commonly used in contrastive learning. For pre-training, this pipeline uses cosine similarity to learn the general trends in the data. Then, it uses ranking similarity in the fine-tuning stage so the model can learn a finer differentiation between waveform details to achieve higher accuracy. There are three main contributions: First, they propose a platform filtering method and self-adaptive FIR filter for data pre-processing, which improves the quality of the data. Then, they propose a new ranking similarity for fine-tuning models. Finally, they create a time series classification model with CNNs to remove high-dimensional features. The proposed method was evaluated on 4 groups of 8 real datasets and compared to several state-of-the-art models.

**Strengths:**

+ The Methodology sections 4.4 and 4.5 are written clearly enough. This paper builds off a previous work, and it is clear which parts of the model were changed. The paper identifies and explains each part that was changed in this updated model pipeline.
+ Section 5.3’s Fine-tuning Analysis is strong. This section explains why certain baselines do well on some datasets and struggle in others. This supports the values reported in Table 2 and make the results more convincing.
+ Figure 3 in section Sorted Similarity Representation is helpful for understanding the impact of the proposed ranking similarity in the pipeline. It provides a visual cluster of the data after using the ranking similarity to show that ranking helped separate the data. The corresponding text also supports this Figure 3 and makes this work stronger.

**Weaknesses:**

- The paper is difficult to read because of poor grammar. Too many sentences are hard to parse.
- For the Methodology section 4.2 and 4.3, the data filtering algorithms are very briefly described, and some pseudocode is provided. It would be easier to understand the algorithms with some more details in the description. For instance, section 4.2 mentions “We use a FIR filter with a low-pass frequency designed to be adaptive. The value is set based on the maximum frequency of the current curve multiplied by sqrt(2)/2”. Although it is explained in the Appendix, a bit more explanation about what makes the filter adaptive and why the values sqrt(2)/2 is used would make this section in the main text more understandable.
- Some of the variables in the algorithms are not explained well. For example, section 4.3 mentions that “During the data platform filtering process, we need to make a reasonable division of the winscale value, otherwise overfiltering may occur”. An explanation of what the winscale value represents and a range of reasonable values would make this section stronger.
- The paper lists 8 datasets used in this work for evaluation and gives a description of each, but there is not much explanation for why these particular datasets were chosen. More detail about why these particular datasets were chosen would help this section. In the methodology these 8 datasets are grouped into pairs where one dataset is used for pre-training and another is used for fine-tuning. After reading the Table 1, it becomes clear that these datasets are paired by similar domains. However, it would be better if this was stated directly in the text instead of implied by the Table 1.
- The Ablation Study and its corresponding Figure 4 are not clear. It states that 7 combinations of DPTSC were compared, but it is not clear what these 7 combinations are, even after re-reading sections of the paper and appendix. The Figure 4 is also harder to read and understand compared to the other figures and tables in this work.

**Questions:**

Please see the strengths and weaknesses. In addition to grammar, the presentation should also be improved. Related Work lists previous papers and their limitations, rather than providing the reader with background information for this paper’s research field. The Ablation Study is vague, and it is unclear what it is comparing. This makes it difficult to see exactly which components in their pipeline are contributing to the improved results.

---

> ### Author Response · Authors · 2023-11-13
> **Reply to PC6z**
>
> First of all, thank you very much for reviewing our work and providing valuable suggestions, as well as recognizing some of our work. We will continue to work hard on my research. Here is my response to the questions you raised.
>
> Q1:
> We will improve the grammar and expression of the article, my English is really not good, thank you for your criticism!
>
> Q2:
> Setting the maximum frequency of $\sqrt{2}/2$ is mainly considered to be the golden ratio, using this value can remove most high-frequency noise, of course, in the actual situation, the use of other values is sometimes necessary.
>
> Q3:
> Winscale is the size of a sliding window, the original text is on page 4 under Algo1.
>
> “First, the algorithm calculates the maximum and minimum amplitudes in the data, and sets the sliding window size to half of the range. Based on the difference between the maximum and minimum amplitudes within the window, the algorithm determines whether to filter the current window.”
>
> For the setting of winscale, the size of winscale is determined by half of the difference between the largest value and the smallest value of the time series, and half of the maximum and minimum value of the amplitude of the time series itself is mainly considered as the size of the sliding window. When the vibration amplitude of the time series is large, it indicates that there is not much of the waveform platform. At this time, we set a larger sliding window size. When the amplitude of the time series is small, it indicates that the probability of the waveform platform part being large becomes more and more, and then we set a smaller sliding window size.
>
> Q4:
> Yes, we quite agree with your suggestion and we will include a note in the text to show that the 8 sets of data are similar in the field.
>
> Q5:
> Figure 4 shows the data preprocessing methods used in this paper, including platform filtering -D, FIR filtering -F and similarity modification -S, which were added or deleted for ablation experiments respectively. The original text reads as follows: “We compared 7 combinations of DPTSC, DPTSC-D, DPTSC-F, DPTSC-S, DPTSC-DF, DPTSCDS, and DPTSC-FS, and DPTSC-DFS by using data platform filtering as condition D, filtering as condition F, and sort similarity as condition S.”
>
> The ablation experiment is a comparison of the previous data preprocessing methods. Finally, thank you again for your advice on our work.

---

### Meta-Review · Area_Chair_1Pzd · 2023-12-07

**Metareview:**

The presentation of this work needs to be significantly improved. Details of the proposed method are missing. Experimental results are not clear or convincing. The technical contributions and novelty are not clear.

**Justification For Why Not Higher Score:**

There are many major concerns raised by the reviewers.

**Justification For Why Not Lower Score:**

N/A

---

### Decision · Program_Chairs · 2024-01-16

Reject